# The Analysis of Arterial Stiffness in Heart Failure Patients: The Prognostic Role of Pulse Wave Velocity, Augmentation Index and Stiffness Index

**DOI:** 10.3390/jcm11123507

**Published:** 2022-06-17

**Authors:** Fabio Anastasio, Marzia Testa, Cinzia Ferreri, Arianna Rossi, Gaetano Ruocco, Mauro Feola

**Affiliations:** 1Cardiology Division, Regina Montis Regalis Hospital, 12084 Cuneo, Italy; fabioanastasio@hotmail.com (F.A.); marzia.testa@aslcn1.it (M.T.); 2Department of Geriatry, University of Turin, 10123 Turin, Italy; cinzia.ferreri@unito.it (C.F.); arirossi_ary@hotmail.com (A.R.); 3Cardiology Division, Ospedali Riuniti di Valdichiana Montepulciano, 53100 Siena, Italy; gmruocco@virgilio.it

**Keywords:** arterial stiffness, heart failure, prognosis, pulse wave velocity, augmentation index and stiffness index

## Abstract

Background: The role of arterial stiffness in the pathogenesis and clinical outcome of heart failure (HF) patients has to be clarified. The aim of this study was to evaluate the prognostic role of arterial stiffness in HF patients discharged after acute episode of decompensation by evaluating cut-off values for clinical assessment. Methods: Patients admitted for decompensated heart failure (ADHF) underwent pre-discharge evaluation. Arterial stiffness was measured by aortic pulse wave velocity (aPWV), augmentation index (AIx75) and stiffness index (β_0_). Patients were also evaluated after discharge for a variable follow-up time. Results: We observed 199 patients (male 61.3%, age 76.2 ± 10.7 years) after discharge for a median of 437 days (IQR 247-903), 69 (34.7%) patients suffered HF with preserved ejection fraction (HFpEF), 45 (22.6%) patients experienced HF with mid-range ejection fraction (HFmEF) and 85 (42.7%) reported an HF with reduced ejection fraction (HFrEF). After the adjustment for principal confounders, aPWV, AIx75 and β_0_ were inversely correlated with free-event survival (*p* = 0.006, *p* < 0.001, *p* = 0.001): only β_0_ was inversely correlated with overall survival (*p* = 0.03). Analysing the threshold, overall survival was inversely correlated with β_0_ ≥3 (HR 2.1, *p* = 0.04) and free-event survival was inversely correlated with aPWV ≥10 m/s (HR 1.7, *p* = 0.03), AIx75 ≥ 25 (HR 2.4, *p* < 0.001), and β_0_ ≥ 3 (HR 2.0, *p* = 0.009). Dividing HF patients for LV ejection fraction, β_0_ and AIx75 appeared to be accurate prognostic predictors among the three different classes according to free-event survival. Conclusions: The non-invasive measurements of arterial stiffness proved to be strong prognostic parameters in HF patients discharged after an acute HF decompensation.

## 1. Introduction

Heart failure (HF) is a clinical syndrome with high prevalence in elderly people. Clinically, HF is classified based on left ventricular (LV) systolic function: HF with preserved ejection fraction (HFpEF) (EF > 50%), HF with mid-range ejection fraction (HFmEF) (40–49%) and HF with reduced ejection fraction (HFrEF) (EF < 40%) [1]. These three classes are characterized by different pathological mechanisms underlying the HF. Indeed, if in HFrEF patients the key is the reduced LV pumping function, the LV diastolic dysfunction plays an essential pathophysiological role in the development of HFpEF. In these patients, an abnormal increase in large arterial stiffness increases the workload on the heart and deteriorates ventricular compliance and ventricular–arterial coupling [2,3,4], leading to the progression of cardiac dysfunction [5,6,7]. Actually, in previous studies, aortic Pulse Wave Velocity (aPWV), the gold-standard non-invasive parameter for studying the large arterial stiffness, proved to be a strong independent predictor of atherosclerotic cardiovascular disease (CVD) and cardiovascular events in patients with preserved LV systolic function [8,9,10,11,12,13]. However, the association between vascular remodelling and increased aPWV was associated with incident HFrEF in age- and sex-adjusted models in a Framingham study population [14,15,16]. On the other side, the association of aPWV with cardiovascular outcomes in HFrEF patients remains uncertain [17]. In these patients, the probabilities of HF-related events were significantly higher in the lower aPWV groups [11,14]. The mechanisms underlying this relation as yet have not been elucidated. This might be explained by the low-pressure and low-velocity pulse waveforms secondary to myocardial loss or degeneration and dysfunction rather than by a real reduction in wave reflections or aortic stiffness in HF [18]. Moreover, the prognostic role of aPWV seems to be unclear in HFmEF patients.

In the 2018 European Society of Cardiology Guidelines, an aPWV threshold of 10 m/s was reported as suggestive of significant alterations of aortic function and of increased mortality and cardiovascular risk in middLe-aged patients with arterial hypertension [18,19,20]. Aortic PWV threshold in older and HF patients was never investigated.

The aim of this observational single-centre study was to analyse the prognostic role of arterial stiffness parameters (aPWV and stiffness index β_0_) and wave reflections (Augmentation Index—AIx75) in HF patients at discharge after an acute episode of decompensation and to elucidate a cut-off of those parameters with good sensibility and specificity for overall survival and free-event survival (all cause of death and rehospitalisation for HF) at a mid-term follow-up.

## 2. Methods

Consecutive HF patients hospitalized for an acute episode of decompensation (de novo or in chronic HF) in Mondovì, Piemonte, Italy, between April 2017 and March 2021, were enrolled. Symptomatic HF diagnosis was defined according to the Framingham criteria [21]. Patients with acute coronary syndrome and those who were receiving dialysis were excluded due to possible bias resulting from an acute haemodynamic condition, as well as impossibility in determining aortic stiffness (lack of quality signal, lack of compliance) or due to absence of therapeutic compliance (assessed on subsequent visits or through phone calls) as well as absence of clinical or by-phone acceptance of a long-term follow-up. The two main endpoints were overall survival and free-event survival. These events were defined as death for any cause or re-hospitalization for acute HF decompensation. Cardiovascular events were evaluated in the follow-up performed by direct clinical examination, through phone calls or through National Institute of Statistics (ISTAT) death records. All patients signed written consent and the study was previously approved by our institutional review board (n.10–18 in 2018).

We investigated patients’ 2019 backgrounds, including age, gender, New York Heart Association (NYHA) functional class, vital signs on admission, comorbidities, laboratory data, and echocardiographic data during hospitalization. Clinical parameters and vitals parameters were measured in patients with stable HF in optimal medical therapy immediately before discharging. Echocardiography was performed using a GE Vivid 7 Pro (General Electric, Boston, MA, USA), according to the recommendations of the American Society of Echocardiography [22]. Left-ventricular systolic dysfunction was defined as an LVEF < 50% calculated by a modified Simpson’s method using biplane apical (2- and 4-chamber) views. The pulmonary artery pressure (PAP) was obtained by determining the peak velocity of the tricuspid regurgitation jet, plus 5 or 10 mmHg for right atrial pressure according to right atrial size, severity of regurgitation and appearance of the inferior vena cava. From Doppler tissue imaging of the annulus, the E′ wave (early annular velocity opposites in direction to the mitral inflow) was determined and the ratio E/E′ calculated. Right ventricular function was investigated by M-mode echocardiography, obtaining the tricuspid annular plane systolic excursion (TAPSE).

Arterial stiffness parameters were evaluated using SphygmoCor XCEL (AtCor Medical, Itasca, IL, US), a non-invasive diagnostic tool for the clinical evaluation of central arterial pressure [23]. The SphygmoCor XCEL System derives the central wave-shaped aortic pressure from the pulsations of the brachial artery cuffs. Waveform analysis provides key parameters that include central systolic and diastolic pressure. The velocity of the arterial pulse wave is detected by the carotid and femoral arterial impulses simultaneously measured in a non-invasive manner (aPWV) [20]. The carotid pulse is measured through the tonometer, whereas the femoral pulse is measured through the pulsations with a cuff placed around the thigh. AIx75 was measured at the level of the carotid artery by obtaining ten high quality pulse wave measuremsents with automatic calculation using the manufacturer’s proprietary software and after normalizing to a heart rate of 75 beats per minute. AIx75 represents the pressure boost that is induced by the return of the reflected waves at the aorta. The stiffness index *β*_0_ was estimated from measurements in the supine position, using the following Equation [18]
$$\beta_{0}=\frac{2*\rho *PWV^{2}}{Pd}-\frac{\ln Pd}{Pref}$$
with *ρ* the blood mass density, taken to be 1050 kg/m^3^, *PWV* the measured aPWV, *Pd* the central DBP, and *Pref* = 100 mm Hg a reference pressure.

### Statistical Analysis

Continuous variables were expressed as mean ± standard deviation (SD) or median and inter-quartile range (IQR) and the categorical variables were presented as absolute value and percentage. Differences between groups were assessed by ANOVA, median test and Kruskal–Wallis test. Chi-square statistics were used to assess differences between categorical variables. Pearson’s correlation coefficient and Cox logistic regression were used to study univariate and multivariate relations between variables and between 1-year overall survival and 1-year free-event survival. Log rank test, Kaplan–Meier and Cox regression were used for univariate and multivariate time-depending analysis such as overall survival and free-event survival. We considered the following to be potential confounding factors that are known to affect aPWV in HF patients: sex, age, LVEF, HR, mean blood pressure (MBP), pro-BNP, serum creatinine, Hb1Ac. Statistical analysis was carried out in an SPSS V.26 statistical software package (SPSS for Windows V26, SPSS Inc., Chicago, IL, USA) and a *p*-value of 0.05 or less was considered statistically significant.

## 3. Results

The analysis included 199 patients: 122 male (61.3%) and 77 female (38.7%). Mean age was 76.2 years ± 10.7 years and 69 (34.7%) patients suffered of HFpEF, 45 (22.6%) patients showed HFmEF and 85 (42.7%) patients were HFrEF. The prevalence of comorbidities and clinical characteristics were showed in Table 1. In total, 15.9% of HFpEF patients were treated with angiotensin receptor/neprilysin inhibitor (ARNI) because of a previous history of HF with ejection fraction <40% that improved their ventricular function and, of course, maintained the optimized therapy. In a median follow-up of 437 days (IQR 247–903), 67 (33.7%) HF patients died, 107 (53.8%) patients showed a combined cardiovascular event in the follow-up (death for any cause or HF re-hospitalization). One-year free-event survival prevalence was 45.2% (71 of 157 patients observed for at least 1 year). No differences were noted between different HF classes (Table 1).

In the whole population, after correction for the major confounder [sex, age, LVEF, HR, MBP, BNP, serum creatinine, Hb1ac], aPWV proved to be an independent factor associated with free-event survival (β 1.13, IC95% 1.04–1.23, *p* = 0.005), failing to demonstrate a significant correlation with the overall survival (*p* = 0.08).

Different values of aPWV were analysed creating thresholds to identify a significant prognostic marker. An aPWV cut-off point of 10 m/s resulted the best indicator of mortality and HF rehospitalisation with a sensibility of 76% and specificity of 46% for overall survival (ROC area 0.61, *p* = 0.009) and with a sensibility of 73% and specificity of 51% for free-event survival (ROC area 0.62, *p* = 0.003). In the entire HF population, the overall survival and free-event survival in patients with aPWV ≥10 m/s were 709 ± 43 and 446 ± 44 days against 922 ± 45 days (*p* = 0.003) and 738 ± 56 days (*p* < 0.001) in patients with aPWV <10 m/s (Figure 1). After the correction for confounders, only free-event survival was significant correlated to aPWV threshold (HR 1.7, 95%CI 1.1–2.7, *p* = 0.03) (Figure 2).

Similar to aPWV, AIx75 proved to be, in the whole population, an independent factor associated with free-event survival (adjusted β 1.02, IC95% 1.01–1.03, *p* < 0.001) but not for overall survival (*p* = 0.29). A cut-off of AIx75 =25 appeared to be the best compromise between sensibility and specificity of 63% and 50% for overall survival (ROC area 0.56, *p* = 0.04) and 65% and 59% for free-event survival (ROC area 0.62, *p* = 0.003). The overall survival and free-event survival were 733 ± 46 days in patients with AIx75 ≥25 against 858 ± 45 days (*p* = 0.06) in patients with AIx75 < 25, and 427 ± 47 days against 715 ± 51 days (*p* < 0.001), respectively (Figure 1). After correction for confounding factors, the cut-off of AIx75 =25 showed significant correlation only with free-event survival (HR 2.4, 95%CI 1.6–3.8, *p* < 0.001) (Figure 2).

Finally, even β_0_ resulted an independent factor associated with the overall survival (adjusted β 1.16, IC95% 1.01–1.33, *p* = 0.03) and with free-event survival (adjusted β 1.23, IC95% 1.09–1.39, *p* = 0.001).

The best cut-off of β_0_ seemed to be =3 obtaining a sensibility of 87% and a specificity of 37% for overall survival (ROC area 0.62, *p* = 0.003) and a sensibility of 87% and a specificity of 44% for free-event survival (ROC area of 0.63 (*p* = 0.001). Patients with β_0_ ≥ 3 showed an overall survival of 720 ± 40 days against 970 ± 46 days in patients with β_0_ <3 (*p* = 0.001) and a free-event survival of 463 ± 41 days against 805 ± 62 days (*p* < 0.001) (Figure 1). β_0_ ≥ 3 was independent correlated with overall survival (adjusted HR 2.1, IC95% 1.0–4.4, *p* = 0.04) and with free-event survival (adjusted HR 2.0, IC95% 1.1–4.2, *p* = 0.03) (Figure 2).

Due the high variability of heart rate in patients with atrial fibrillation (AF), the analysis was repeated adding AF as confounder and no significant differences in results were detected.

### Sensitivity Analysis

Dividing the population according to the classification of HF, we analysed the overall survival and free-event survival (all cause of death and rehospitalisation for HF) in order to correlate the parameters derived from the arterial stiffness measurements and prognosis in different HF setting. In HFpEF, in multivariate analysis, all causes of death and rehospitalisation were correctly predicted by AIx75 (β 1.05, IC95% 1.01–1.08, *p* = 0.004).

In HFmEF, free-event survival was inversely correlated with aPWV (β 1.28, IC95% 1.04–1.58, *p* = 0.02), AIx75 (β 1.05, IC95% 1.00–1.11, *p* = 0.03), and β_0_ (β 1.43, IC95% 1.05–1.95, *p* = 0.02).

Finally, in patients with HFrEF, overall survival was independently correlated with aPWV (β 1.27, IC95% 1.08–1.50, *p* = 0.005) and β_0_ (β 1.39, IC95% 1.15–1.68, *p* = 0.001). Whereas AIx75 (β 1.02, IC95% 1.00–1.03, *p* = 0.04), and β_0_ (β 1.21, IC95% 1.01–1.47, *p* = 0.04) seemed to be good prognostic parameters of free-event survival.

Using the previous thresholds described, in HFpEF, no correlations were observed between stiffness parameters and overall or free-event survival. In HFmEF, overall survival was independently and inversely correlated with AIx75 = 25 (HR 4.54, IC95% 1.14–18.02, *p* = 0.03) and free-event survival seemed to be independently and inversely correlated with aPWV =10 (HR 7.04, IC95% 1.74–23.93, *p* = 0.006) and AIx75 =2 5 (HR 5.11, IC95% 1.71–16.21, *p* = 0.005) (Figure 2, Figure 3 and Figure 4). In HFrEF patients, only AIx75 = 25 (HR 2.64, IC95% 1.28–5.53, *p* = 0.009) demonstrated a predictive value of free-event survival (Figure 2, Figure 3 and Figure 4).

## 4. Discussion

In our clinical experience, arterial stiffness parameters proved to be a strong prognostic values in HF patients discharged alive after an acute decompensation. All stiffness parameters (aPWV, β_0_) and wave reflection parameters (Aix75) were correlated with cardiovascular events scheduled in the mid-term follow-up (all-cause mortality and re-hospitalization for HF). In particular, unit growth of aPWV, AIx75 or β_0_ was associated with an increase in risk of death for any cause or HF re-hospitalization of 13%, 2% and 23%. β_0_ was the only factor still associated with overall survival, with a rise of 16% in the risk of death for any cause.

On the other hand, using a threshold of 10 m/s, HF patients presenting a more rapid pulse wave velocity seemed to predict a worse prognosis (death/rehospitalisation) with an HR of 1.7 in the discharged HF patients. Moreover, HF patients having a value of AIx75 ≥ 25 experienced CV events 2.4 times higher than those discharged with a value <25. A β_0_ threshold of 3 was associated with an increased risk of 2.1 for death for any cause and a threshold of 2 for scheduled cardiovascular events of death for any cause or HF re-hospitalization.

In the literature, the role of arterial stiffness in HF patients remains under debate. In patients with chronic stable HFpEF a higher value of aPWV has been associated with an increment of hospitalization for HF and CV mortality [14,15,24]. In their clinical review, Weber and Chirinos [25] recently highlighted that central pressure and wave reflections are both related to the left-ventricular late-systolic afterload, ventricular remodeling, diastolic dysfunction and the risk of new-onset HF. The reduced aortic compliance constitutes an increased load on cardiac output which prolong the duration of systole. During prolonged contraction, an early reflected wave represents an additional burden with consequent left-ventricle remodeling and myocardial dysfunction. The effect of the wave reflection on myocardial load is modulated by a contraction pattern and the time course of myocardial wall stress. Left ventricles in which the mid-systolic shift in the pressure-stress relation is impaired (due to a reduced ejection fraction, concentric geometric remodeling and/or reduced early systolic ejection fraction) fail to protect cardiomyocytes against the load induced by wave reflections in late systole, a period of vulnerability for loading. This may represent a vicious cycle that might determine the development and progression of heart failure. Moreover, left ventricular hypertrophy, a marker of organ damage in hypertension, is an important intermediate step from hypertension to development of HF [26] and might be influenced by the time of the wave reflection during the cardiac cycle. In fact, left ventricular mass seemed to be more correlated to pulsatory pressure than to mean arterial pressure [27]. A recent review [28] underlined the strict linkage between arterial stiffness, impaired renal function and development of HF, confirming that the presence of renal failure plays an important role in the development of vascular damage (monitored by aPWV) and HF. On the other hand, in HF patients with reduced systolic function, a lower aPWV may reflect the severity of loss of ventricular function rather than aortic stiffness [18]. This relation can be demonstrated by the observed modest increases in SBP, CO, and aPWV in HFrEF patients after the implantation of CRT, very often associated with favourable clinical outcomes [29]. However, results on the correlation between parameters obtained by the measurement of arterial stiffness and wave reflection and survival in HFrEF patients are currently inconclusive [14,30,31].

Furthermore, in our sensitivity analysis that divides HF patients according to the current ESC guidelines, β_0_ appears to be the best predictor of cardiovascular events in the follow-up. This might be explained by the fact that β_0_ is a structural factor which is independent of the haemodynamic state. On the other hand, in relation to the analysis cut-off, AIx75 = 25 appears to be the best predictor of HF re-hospitalization in all HF categories of systolic function. The better prognostic role of AIx75 might be caused by the effect of an earlier reflected wave on an impaired systolic function resulting in a pressure overload on the left ventricle. These data allow a hypothesis that the increasing in arterial stiffness and an early reflected wave could cause an additional load on the ejection of the left ventricle and could play a role in determining left ventricular remodeling. This might represent a vicious circle that favours the development and progression of heart failure. This study has many limitations. First of all, there is the retrospective nature of the study. Moreover, the sample size might have influenced the sensitivity analysis when a different pattern of HF is considered. Additionally, HF patients were treated with a variety of cardiovascular agents, which may have influenced their aPWV values. Finally, HF patients with lower aPWV values having a low cardiac output were only estimated using echocardiography, rather than by confirmation by invasive/non-invasive measurements of cardiac output. These data have to be further investigated by multicentre trials.

## 5. Conclusions

In conclusion, in our clinical experience, parameters obtained from the measurement of arterial stiffness proved to be predictive of mid-term outcomes in HF patients discharged alive after an acute cardiac decompensation. Aortic PWV, AIx75 and β_0_ were inversely correlated with free-event survival, and conversely, only β_0_ was inversely correlated with overall survival. A value of aPWV ≥10 m/s obtained an HR 1.7 in predicting death or rehospitalisation for HF in the entire HF population. When HF patients are divided by LV ejection fraction, β_0_ and AIx75 appeared to be accurate prognostic predictors among the different class according to the free-event survival.

According to our results, the pre-discharge evaluation of arterial stiffness values seemed to be useful in HF patients and the parameters obtained might be considered to be independent markers of clinical prognosis. Moreover, it can be hypothesized that in HF patients, aPWV, AIx75 and β_0_ might be considered in relation to the determinism and progression of heart failure.

## Figures and Tables

**Figure 1 jcm-11-03507-f001:**
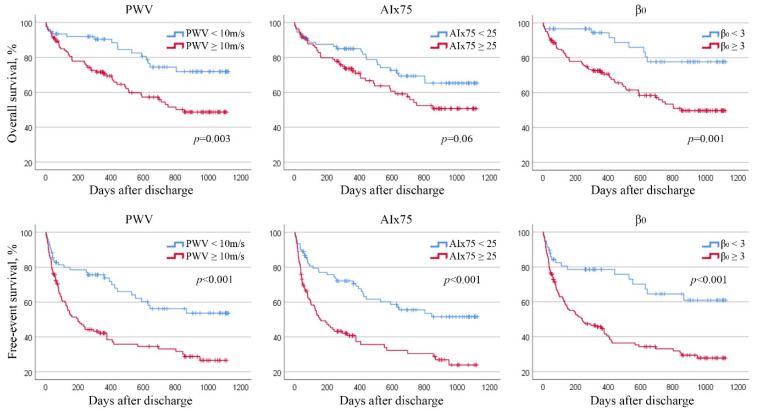
Kaplan–Meier plot for overall survival and free-event survival in patients divided by aPWV, AIx75 and β_0_ thresholds.

**Figure 2 jcm-11-03507-f002:**
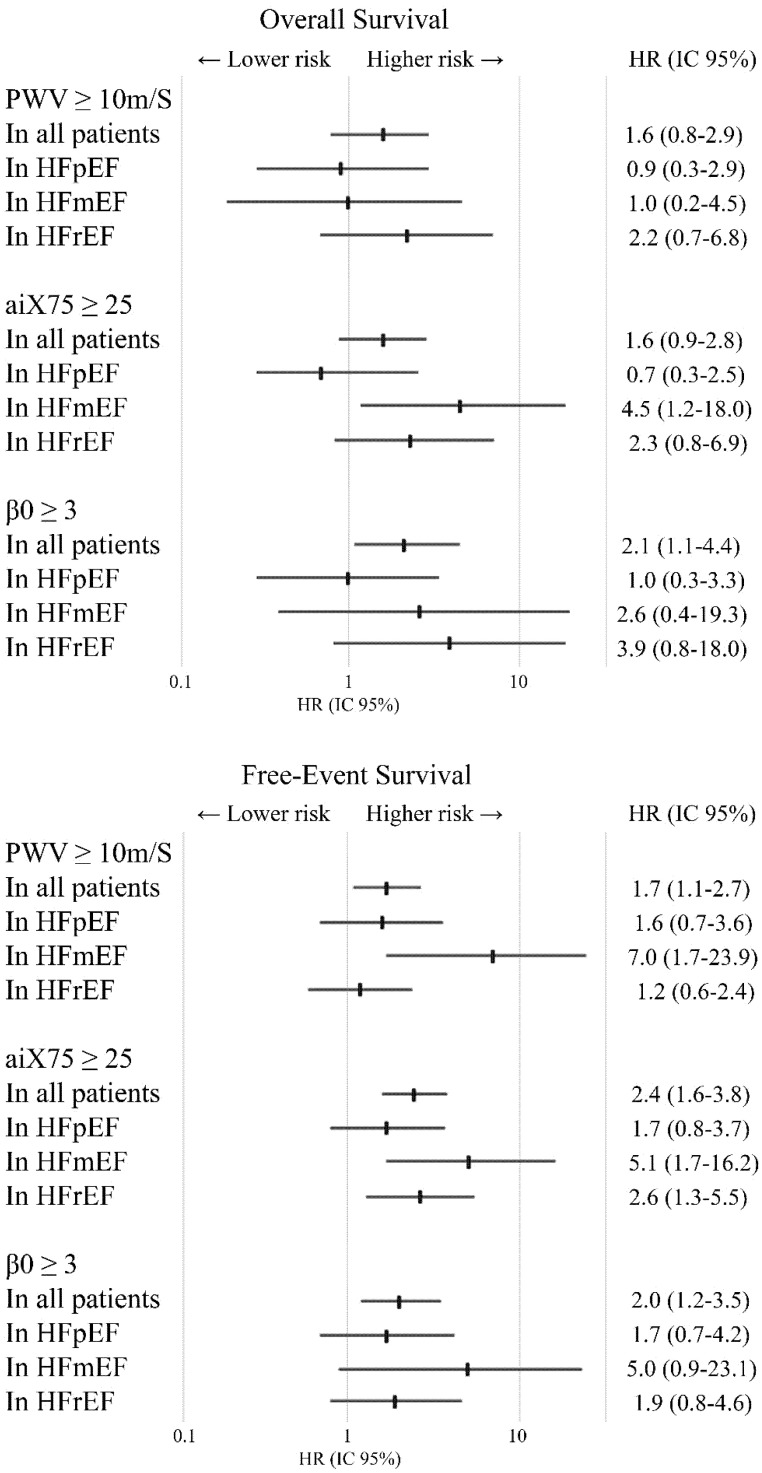
Adjusted HR by multivariate Cox regression for overall survival and free-event survival in patients divided in HFpEF, HFmEF and HFrEF.

**Figure 3 jcm-11-03507-f003:**
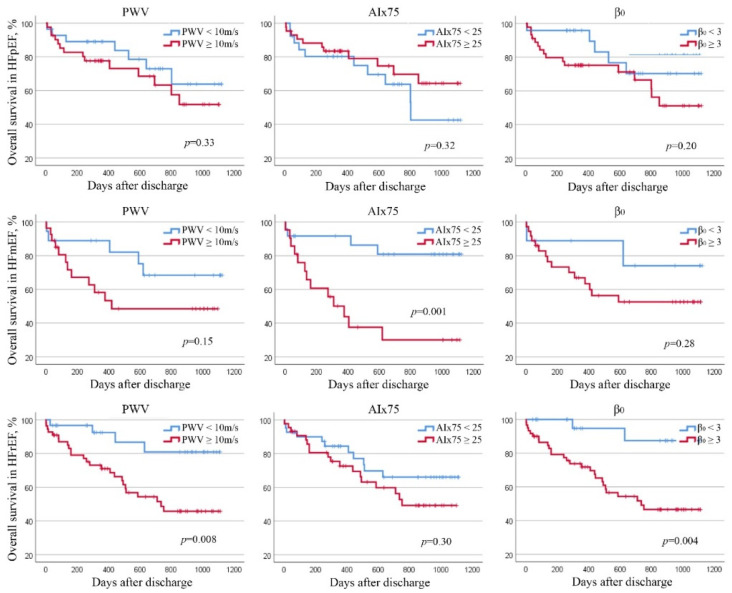
Kaplan–Meier plot for overall survival in patients with HFpEF, HFmEF and HFrEF divided by aPWV, AIx75 and β_0_ thresholds.

**Figure 4 jcm-11-03507-f004:**
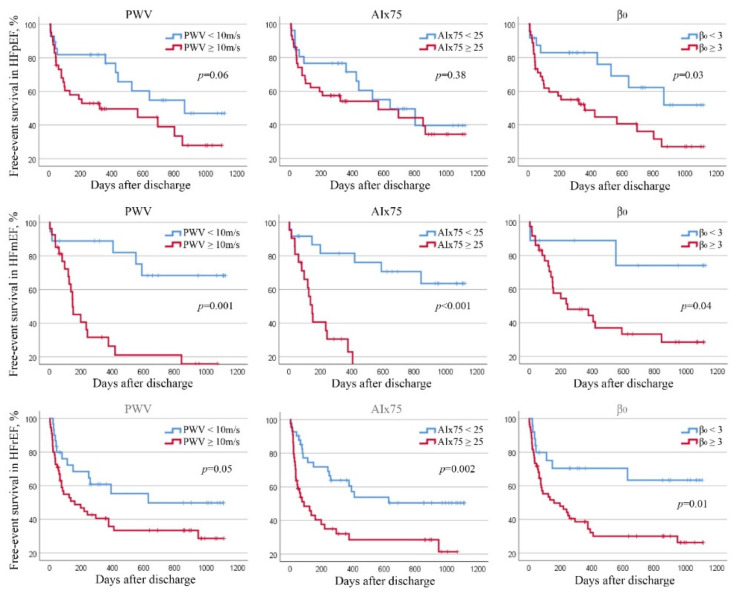
Kaplan–Meier plot for free-event survival in patients with HFpEF, HFmEF and HFrEF divided by aPWV, AIx75 and β_0_ thresholds.

**Table 1 jcm-11-03507-t001:** Basal characteristics of patients.

	HFpEF (*n* = 69)	HFmEF (*n* = 45)	HFrEF (*n* = 85)	*p*
Sex (male)	31 (44.9%) *	30 (66.7%)	61 (71.8%) *	0.002
Age	77.9 ± 9.9 *	75.0 ± 20.0	69.6 ± 18.3 *	0.007
DM	32 (46.4%)	14 (31.1%)	41 (48.2%)	0.15
CAD	8 (11.6%) *	13 (28.9%)	41 (48.9%) *	<0.001
VHD	30 (42.5%)	24 (53.3%)	39 (45.9%)	0.58
AF	14 (20.3%)	7 (15.6%)	13 (15.3%)	0.93
Hypertension	28 (40.6%)	10 (22.2%)	20 (23.5%)	0.15
ACE/ARB	38/69 (55.1%)	19/45 (42.2%)	34/85 (40.0%)	0.42
ARNI/ARB	11/69 (15.9%)	2/45 (4.4%)	17/85 (20.0%)	<0.001
BB	57/69 (82.6%)	41/45 (91.1%)	79/85 (92.9%)	0.11
Ivabradine	6/69 (8.7%)	4/45 (8.8%)	10/85 (11.8%)	0.69
MRA	53/69 (76.8%)	33/45 (73.3%)	74/85 (87.0%)	0.10
Furosemide (mg/die)	75 [46–125]	60 [50–125]	75 [50–195]	0.68
Last 24 h diuresis (mL)	1400 [1100–2000]	1700 [1300–2400]	1600 [1250–2500]	0.12
Weight decrease (kg)	2.5 [1.8–3.2]	2.2 [0.9–5.4]	3.4 [1.3–4.3]	0.35
Weight decrease (%)	3.0 [2.4–4.6]	3.2 [1.5–8.1]	3.9 [1.8–6.0]	0.62
SBP mmHg	128 ± 18	124 ± 19	124 ± 17	0.28
DBP mmHg	75 ± 13	71 ± 14	75 ± 13	0.19
Hb1ac %	6.6 ± 1.3	6.5 ± 0.9	6.7 ± 1.2	0.47
Creatinine mg/dL	1.0 [0.8–1.2] *	1.2 [0.8–1.8]	1.3 [0.0–1.9] *	0.007
Pro-BNP pg/mL	4310 [2118–10,338] *	6855 [2588–15,185]	10,483 [4694–24,388] *	<0.001
LVEF %	55 ± 5 #*	44 ± 3 #φ	28 ± 6 *φ	<0.001
LVEDD mm	46 ± 9 #*	53 ± 15 #φ	62 ± 15 *φ	<0.001
TAPSE mm	20 [18–23]	18 [17–20]	16 [14–19]	<0.001
PAPs mmHg	44.3 ± 20.3	38.5 ± 11.4	40.6 ± 13.9	0.51
aPWV m/s	10.7 ± 2.4	10.9 ± 2.5	11.2 ± 2.9	0.46
AIx75	29.6 ± 13.3	23.4 ± 13.9	27.5 ± 18.1	0.32
β_0_	3.8 ± 1.5	4.2 ± 1.7	4.2 ± 2.2	0.23
Death	21 (30.4%)	17 (37.8%)	29 (34.1%)	0.71
Event	35 (50.7%)	24 (53.3%)	48 (56.5%)	0.77
1-year free-event	24/50 (48.0%)	19/37 (51.4%)	28/70 (40.0%)	0.47

Abbreviations: DM diabetes mellitus, CAD coronary arteries disease, VHD valvular heart disease, AF atrial fibrillation, ACE angiotensin-converting enzyme, ARB angiotensin-receptor blocker, BB beta-blocker, ARNI angiotensin receptor/neprilysin inhibitor, MRA mineralocorticoid receptor antagonists, SDB systolic blood pressure, DBP diastolic blood pressure, BNP brain natriuretic peptide, LVEF left ventricular ejection fraction, LVEDD left ventricular end-diastolic diameter, aPWV carotid-femoral pulse wave velocity. Data are show as absolute number (%) or mean (±standard deviation) or median [interquartile]. # Significant difference between group 1 and 2, * Significant difference between group 1 and 3, φ significant difference between group 2 and 3.

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
