# Peer review of "The Analysis of Arterial Stiffness in Heart Failure Patients: The Prognostic Role of Pulse Wave Velocity, Augmentation Index and Stiffness Index"

_jcm, 2022, doi:10.3390/jcm11123507_

Round 1

Reviewer 1 Report

the authors addressed all the queries indicated in the previous revision. Good job, nice data, pretty manuscrit

Author Response

thank you

Reviewer 2 Report

The analysis of Arterial Stiffness in heart failure patients: the prognostic role of pulse wave velocity, augmentation index and stiffness index.

Thanks a lot for giving me the opportunity to serve MDPI

1.     Methods: Lines 96-97; explain why acute coronary syndrome and dialysis patients were excluded. Also explain what you mean by impossibility in determining PWV, and how the investigators assessed therapeutic compliance

2.     Always refer to PWV as aPWV (aortic) unless it represent another region of the arterial tree

3.     Results: line 172 and 184, please say associated with free-event survival

4.     Sensitivity analysis is better phrased as;

In HFpEF, the following stiffness parameters were found to be associated with overall survival or free-event survival

In HFmrEF, the following …..

In HFrEF, the following …..

5.     Discussion to be started with the most important findings of the current study (not putting it in lines 251-263)

Author Response

thank you

This manuscript is a resubmission of an earlier submission. The following is a list of the peer review reports and author responses from that submission.

Round 1

Reviewer 1 Report

The analysis of Arterial Stiffness in heart failure patients: the prognostic role of pulse wave velocity, augmentation index and stiffness index.

Thanks a lot for giving me the opportunity to serve MDPI

I read with interest the above paper and I have the following suggestions;

Overall:

  1. The manuscript needs proofreading to improve the language

Introduction:

  1. Line 57: the association of PWV with cardiovascular outcomes in HFrEF patients remains uncertain. I suggest replacing it with “increased aortic PWV was associated with incident HFrEF in age- and sex-adjusted models in Framingham study population”.
  2. Reference 19 seems inserted by mistake “Descriptive analysis of long COVID sequela identified in a multidisciplinary clinic serving hospitalized and non-hospitalized patients”
  3. Line 58-62: Analysis based of brachial-ankle wave velocity cannot be compared with results of aortic pPWV. They reflect a different part of the arterial tree. It is better to use aortic PWV only
  4. Line 63-65: clarify if it was aortic PWV or not?

Methods:

  1. The authors need to clarify if this was the first decompensation to present with? Or for how long these patients have had heart failure?
  2. Line 73: remove “well-trained”. Any cardiologist shall be knowledgeable to diagnose HF
  3. Exclusion criteria need clarification; why patients with acute coronary syndrome were excluded? especially that arterial stiffness assessment took place after stabilization and pre-discharge/ patients on dialysis are a high-risk group, why they were excluded/what do you mean by impossibility in determining aortic stiffness?/ absence of therapeutic compliance is very weird reason for exclusion. How it was defined? and how it was identified?
  4. Spelling mistakes in line 78 “through”
  5. If clinical parameters were taken pre-discharge, vital signs used to derive the arterial stiffness indices from, should be pre-discharge, not on admission (line 80)
  6. Line 83: again remove “expert”
  7. Confounding factors: why the authors selected mean blood pressure, while SBP is more important
  8. IRB approval and consent process not mentioned

Results:

  1. Please refer to PWV as aPWV
  2. Figures 1-4 need P values to be added. Also, Figure 2, lacks legends to show which is overall survival and which is event-free survival
  3. Sensitivity analysis needs to be re-written to give the most important message

Discussion

  1. Please start with the most important findings of your study “ in points” similar to what was mentioned in lines190-194, then start to discuss them and emphasize what they add to the current literature
  2. It is also useful to compare arterial stiffness results with those in other populations with heart failure as this is a new risk factor that is being characterized in heart failure population, such as in; Journal of International Medical Research 48(4) 1–15

Limitation:

  1. The excluded patients need to come here, as they represent a big sample of those at risk of events and mortality

Author Response

Dear  Editor,

please receive 1 electronic copy of our  revised manuscript entitled The analysis of Arterial Stiffness in heart failure patients: the prognostic role of pulse wave velocity, augmentation index and stiffness index be reconsidered for publication in your prestigious Journal.  All the changes in the manuscript has been evidenced .

Answers to Reviewer 1.

The authors appreciated the revision that ameliorated the quality of the manuscript.

Answers to REV 1.

1) changed as suggested.

2) Removed

3) Removed the analysis based on brachial-ankle wave velocity

4) Clarified in line 63-65

Methods.

1) We precised if it was a rehospitalization for HF or de-novo presentation

2) removed

3) we defined well the absence of therapeutic compliance

4) changed

5) modified as adviced

6) removed

7)

8) Mentioned

Results.

1)     We preferred PWV because of the most important articles referring of arterial stiffness defined pulse wave velocity as PWV.

2)     2) a p-value has been added as requested.

3)     We tried to re-write the ‘sensitivity analysis’ as adviced.

Discussion.

We corrected the capther and wrote a ‘conclusion’ with a more conclusive issues. 

Answers to reviewer 2.

The authors thank for the advices  that improved the manuscript.

1. we apologized for the mistakes in the figures that was changed and clarified. 

2. We explained why 15.9% of the preserved HF patients had in their therapy ARNI in Results.  

3.We recalculated the statistical analysis according to the presence of atrial fibrillation as adviced.

4.We created a new paragraph as adviced.

5. The manuscript has been revised by an expert native language revisor in order to improve the fluency.   

I  hope that you will find our work of interest and will be looking forward to receiving your comments.

Please do not hesitate to contact me should you  require any further information.

Mondovi’ 07/5/2022                                                                 Dr. Feola Mauro, MD FESC

Reviewer 2 Report

the authors presented data on the vascular stiffness in heart failure. The aim is worthy of interest and introduction makes data may be really appealing for readers and clinicians. Methods are clearly described. Results are clearly presented and well discussed.

However, there are some major issues that makes the manuscript disappointing when a deep and comprehensive analysis was done. These queries need to be addressed to ensure the high quality of research.

1- there is no correlation between data described in main text and figures provided to support it. In fact, data showed in figure appears the exact opposite of those presented in main text. This is a really major issue because it is not clear what are the correct data a reader should appreciate and consider.

2- ARNI should be given only in HFrEF. On the contrary, in table 1 it is presented the evaluation of patients according to ejection fraction. Looking to concomitant treatment, about 15% patients with HFpEF are treated with ARNI. Moreover, there is a significant difference in the comparison between the three groups, I guess in an ANOVA evaluation. How is it possible?
If these patients were on ARNI treatment at the time of evaluation/enrollment, I suggest to exclude them or consider as HFrEF, especially if an echocardiogram was available in patients' history.

3- about 10-20% of patients were affected by atrial fibrillation. Pulse wave velocity may be affected by the beat-to-beat change in cardiac output. Therefore, I suggest to perform a multivariate analysis not only for heart rate but also according to the presence or not of atrial fibrillation

4- there is not a real conclusion paragraph. I suggest to include one

5- some point, in particular in results, are not easy to be understood. I suggest a native language revision to increase the English fluency

Author Response

(The authors gave the same response as above.)
